# Low Genetic Variability in *Bemisia tabaci* MEAM1 Populations within Farmscapes of Georgia, USA

**DOI:** 10.3390/insects11120834

**Published:** 2020-11-26

**Authors:** Saurabh Gautam, Michael S. Crossley, Bhabesh Dutta, Timothy Coolong, Alvin M. Simmons, Andre da Silva, William E. Snyder, Rajagopalbabu Srinivasan

**Affiliations:** 1Department of Entomology, University of Georgia, 1109 Experiment Street, Griffin, GA 30223, USA; sg37721@uga.edu; 2Department of Entomology, University of Georgia, 120 Cedar St., 413 Bioscience Bldg., Athens, GA 30602, USA; msc88000@uga.edu (M.S.C.); wesnyder@uga.edu (W.E.S.); 3Department of Plant Pathology, University of Georgia, 3250 Rainwater Road, Tifton, GA 31793, USA; bhabesh@uga.edu; 4Department of Horticulture, University of Georgia, 3250 Rainwater Road, Tifton, GA 31793, USA; tcoolong@uga.edu (T.C.); adasilva@uga.edu (A.d.S.); 5U.S. Vegetable Laboratory, Agricultural Research Service, USDA, 2700 Savannah Hwy, Charleston, SC 29414, USA; alvin.simmons@usda.gov

**Keywords:** *Bemisia tabaci*, farmscape, genetic diversity, microsatellite markers, population genetics

## Abstract

**Simple Summary:**

Sweetpotato whitefly, *Bemisia tabaci* Gennadius, is a serious pest of many agricultural crops worldwide. Numerous studies have examined the genetic structure of whitefly populations separated by geographical barriers; however, very few have assessed the population structure of *B. tabaci* at a farmscape level. A farmscape in this study is defined as heterogenous habitat with crop and non-crop areas spanning approximately 8 square kilometers. To assess the roles of farmscapes as drivers of *B. tabaci* genetic variation, thirty-five populations of the sweetpotato whitefly were collected from crop and non-crop plant species from fifteen farmscapes. Using mitochondrial COI gene sequences (mtCOI) and six nuclear microsatellite markers, the genetic diversity and genetic differentiation among collected *B. tabaci* MEAM1 populations were examined. Haplotype analysis using mtCOI sequences revealed the presence of a single *B. tabaci* MEAM1 haplotype across farmscapes of Georgia. Results from microsatellite markers further showed no significant genetic structuring among populations that corresponded to plant species or farmscapes from which they were collected. Annual whitefly population explosions and subsequent dispersal might have facilitated the persistence of a single panmictic *B. tabaci* population over all sampled farmscapes in this region.

**Abstract:**

*Bemisia tabaci* is a whitefly species complex comprising important phloem feeding insect pests and plant virus vectors of many agricultural crops. Middle East–Asia Minor 1 (MEAM1) and Mediterranean (MED) are the two most invasive members of the *B. tabaci* species complex worldwide. The diversity of agroecosystems invaded by *B. tabaci* could potentially influence their population structure, but this has not been assessed at a farmscape level. A farmscape in this study is defined as heterogenous habitat with crop and non-crop areas spanning ~8 square kilometers. In this study, mitochondrial COI gene (mtCOI) sequences and six microsatellite markers were used to examine the population structure of *B. tabaci* MEAM1 colonizing different plant species at a farmscape level in Georgia, United States. Thirty-five populations of adult whiteflies on row and vegetable crops and weeds across major agricultural regions of Georgia were collected from fifteen farmscapes. Based on morphological features and mtCOI sequences, five species/cryptic species of whiteflies (*B. tabaci* MEAM1, *B. tabaci* MED, *Dialeurodes citri*, *Trialeurodes abutiloneus*, *T. vaporariorum*) were found. Analysis of 102 mtCOI sequences revealed the presence of a single *B. tabaci* MEAM1 haplotype across farmscapes in Georgia. Population genetics analyses (AMOVA, PCA and STRUCTURE) of *B. tabaci* MEAM1 (microsatellite data) revealed only minimal genetic differences among collected populations within and among farmscapes. Overall, our results suggest that there is a high level of gene flow among *B. tabaci* MEAM1 populations among farmscapes in Georgia. Frequent whitefly population explosions driven by a single or a few major whitefly-suitable hosts planted on a wide spatial scale may be the key factor behind the persistence of a single panmictic population over Georgia’s farmscapes. These population structuring effects are useful for delineating the spatial scale at which whiteflies must be managed and predicting the speed at which alleles associated with insecticide resistance might spread.

## 1. Introduction

Insect herbivores rely on living plants for food and habitat. Accordingly, host plants are among the most important ecological factors that drive genetic diversity within and among insect herbivore populations [1]. Insect herbivores are also exposed to selection pressure from several factors such as agricultural practices including spatial and temporal cropping patterns and insecticide usage that can influence insect population genetics [2,3,4,5,6,7,8]. Together, these selective forces may act at the level of a “farmscape”, which is a heterogenous habitat with crop and non-crop areas that herbivores can relatively easily move between [2]. For instance, farmscapes defined by a single ephemeral crop and managed with similar agricultural practices may favor selection for a narrow range of herbivore traits that closely match those relatively homogeneous conditions. On the other hand, very diverse farmscapes that include many crop and non-crop plant species managed using different practices can encourage the maintenance of genetically diverse herbivore populations with the broad variety of traits needed to exploit different habitats; this may be particularly true for polyphagous herbivores capable of exploiting many different host plant species. In either case, selective pressures at the farmscape level, coupled with reproductive isolation, can result in the development of host- or farmscape-associated genetic differentiation [3,4] and landscape/farmscape-associated populations [5,6,7].

Sweetpotato whitefly, *Bemisia tabaci* Gennadius (Hemiptera: Aleyrodidae) is a serious pest of open-field crop production systems throughout the world. Nymphs and adults of *B. tabaci* are phloem feeders and are typically found on the abaxial leaf surfaces of their hosts [8,9]. Their direct feeding causes phytotoxic effects to crops such as silvering in leaves of squash (*Cucurbita pepo* L.) and irregular fruit ripening in tomato (*Solanum lycopersicum* L.) [10,11,12]. Aside from causing direct feeding damage, *B. tabaci* transmits multiple plant-pathogenic viruses to important crops [13,14,15]. *Bemisia tabaci* is a species complex that encompasses more than 40 cryptic species [16,17,18]. Middle East–Asia Minor 1 (MEAM1, formerly known as the B biotype) and Mediterranean (MED, formerly known as the Q biotype) are the two most invasive members of *B. tabaci* worldwide [19,20]. MEAM1 cryptic species of *B. tabaci* was first reported in the United States in the mid-1980s, and has since become the predominant cryptic species in the country [21,22,23]. It readily colonizes squash, watermelon (*Citrullus lanatus* L.), cantaloupe (*Cucumis melo* L.), tomato, snap bean (*Phaseolus vulgaris* L.), and other vegetable crops, while transmitting a wide range of plant viruses in the southwestern and southeastern United States [24]. In 2004, *B. tabaci* MED was first documented on poinsettia (*Euphorbia pulcherrima* Willd. ex Klotsch) in Arizona [25]. Since then, *B. tabaci* MED has rapidly spread in the United states, but is restricted primarily to greenhouse-grown ornamentals [26]. 

Numerous studies have examined the population genetics of *B. tabaci* at broad spatial scales [27,28,29,30,31,32], but less is known about how the genetic differentiation and diversity of *B. tabaci* populations vary at a farmscape level, i.e., between spatially adjacent crop and non-crop habitats. Variability in host-plant resistance, cropping patterns, landscape composition and configuration, and insecticide application patterns all might alter *B. tabaci* genetic population structure [4,33,34,35]. In turn, documenting low or high rates of gene flow can allow crop managers to determine if whiteflies fall into local or regional populations, so that pest control efforts can be organized at the appropriate scale. The mitochondrial cytochrome oxidase subunit I (mtCOI) gene, which mutates at a rapid rate compared with nuclear genes, has typically been used for studying *B. tabaci* evolutionary patterns and phylogenetic relationships [36,37,38,39]. The partial sequence of mtCOI gene is most effective as a molecular marker for the taxonomy and identification of species within the genus *Bemisia* because of the similarities in the morphology of whitefly adults and pupae of several members within this species complex [19,40]. However, the usefulness of mtCOI as a molecular marker to exclusively identify differences at a population level may be limited by its lack of resolution. Insect herbivores’ population structure could be more effectively examined by microsatellite markers due to their ease of use, high polymorphism, co-dominant inheritance, and even distribution throughout the genome [41]. Many microsatellite markers have been developed for whitefly population genetic analysis, identification of hybrids between cryptic species, determination of insecticide resistance levels among populations, and population structure of *B. tabaci* at a broad scale [26,28,29,30,31,32,42]. 

To assess the roles of host plants and farmscapes as drivers of the population structure of *B. tabaci* in Georgia, we collected whiteflies from 35 populations, 14 plant species, and 15 farmscapes (defined here as an area ~8 square kilometers and at least 16 km apart). Whiteflies were then identified to the species level using a combination of morphology and mtCOI sequencing, and *B. tabaci* populations were grouped into MED and MEAM1 cryptic species [43]. We then used six polymorphic microsatellite markers to determine the population structure of the *B. tabaci* MEAM1 populations in order to determine whether *B. tabaci* MEAM1 populations exhibited genetic differentiation by (1) host plant and/or (2) farmscape. 

## 2. Materials and Methods 

### 2.1. Whitefly Collections

A total of 35 different populations of whiteflies were collected from 15 farmscapes located in 15 different counties of Georgia, USA (Table 1). Here, “population” refers to whiteflies collected from a single plant species, while “farmscape” is a broader area as defined above [44,45]. For every population, approximately 100 whitefly adults were collected from 5 to 10 different plants of the same species that were at least 1 m apart using an aspirator. Samples were stored in 95% ethanol at −80 °C until DNA extraction. For population structure analyses, populations were either grouped by host plants or farmscapes. 

### 2.2. DNA Extraction 

Total DNA was extracted from individual whiteflies using InstaGene Matrix containing six percent Chelex resin (Bio-Rad, Hercules, CA, USA). Individual whiteflies were homogenized in 1 mL of autoclaved distilled water in a 1.5 mL microcentrifuge tube and centrifuged for 1 min at 12,000 rpm. The supernatants were discarded and 50 μL of InstaGene matrix was added to the pellet. Microcentrifuge tubes were then incubated at 56 °C for 20 min and vortexed for 10 s. The tubes were again incubated for 8 min at 100 °C, vortexed for 10 s, and centrifuged at 12,000 rpm for 3 min. Extracted DNA was stored at −20 °C until used. 

### 2.3. Determination of Whitefly Species

Collected whitefly populations were placed in individual 9 cm Petri dishes, and under a dissecting microscope at 20× magnification, individuals were grouped into species using a whitefly identification guide [46]. Species identity was further confirmed by amplifying and sequencing the 5′ end of the mitochondrial DNA barcode region of three representative whiteflies from each group using universal primers LCO1490 (5′-GGTCAACAAATCATAAAGATATTGG-3′) and HCO2198 (5′-TAAACTTCAGGGTGACCAAAAAATCA-3′) [43]. Polymerase chain reaction (PCR) was conducted using 2X GoTaq^®^ Green Master Mix (Promega, Madison, WI, USA) in an Eppendorf Mastercycler^®^ pro thermocycler (Eppendorf, Hamburg, Germany). The 50 μL PCR mixture contained 25 μL of Master Mix, 0.5μM of forward and reverse primers, 20 ng DNA, and nuclease-free water. PCR conditions were 5 min of initial denaturation followed by five cycles of 40 s at 94 °C, 40 s at 45 °C, and 60 s at 72 °C; and then 35 cycles of 40 s at 94 °C, 40 s at 51 °C, and 60 s at 72 °C, and a final extension period of 72 °C for 10 min [47]. Successful amplification was confirmed by running 10 μL of PCR products on 1% agarose gels stained with GelRed (Biotium, Fremont, CA, USA). Remaining PCR products were purified using the GeneJET PCR Purification Kit as per the manufacturer’s instructions (ThermoFisher Scientific, Waltham, MA, USA). Purified PCR products were sequenced using the SimpleSeq Kit (Eurofins Genomics, Louisville, KY, USA), and whitefly species identity was confirmed using Basic Local Alignment Search Tool (BLAST) available at the National Center for Biotechnology Information (NCBI) webpage. 

The mtCOI region amplified by primer pairs LCO1490 and HCO2198 was not effective in differentiating *B. tabaci* cryptic species. Therefore, *B. tabaci* individuals (three per population: 35 × 3 = 105) were further identified to the cryptic species level by amplifying and sequencing 867 bp of the 3′end of mtCOI gene using the primers and conditions described by Mugerwa et al. 2018 [18]. Briefly, 0.5 μM of the primers 2195Bt (5′-TGRTTTTTTGGTCATCCRGAAGT-3′) and C012/Bt-sh2 (5′-TTTACTGCACTTTCTGCC-3′) were combined with 20 ng DNA, 2× GoTaq Green Master Mix, and nuclease-free water to a final reaction volume of 50 μL. PCR was performed in an Eppendorf Mastercycler^®^ pro thermocycler with an initial denaturation at 94 °C for 5 min followed by 40 cycles of 40 s at 94 °C, 40 s at 52 °C, and 60 s at 72 °C, and a final extension period of 72 °C for 10 min. PCR products were purified and sequenced as described above. *Bemisia tabaci* cryptic species determinations were based on direct sequence comparisons using the web based NCBI BLAST sequence comparison application. Whitefly species delimitation was based on 3.50% partial mtCOI gene sequence divergence [19]. *Bemisia tabaci* individuals were designated as MEAM1 or MED based on ≥96.50% mtCOI sequence similarity with the MEAM1 (GenBank accession number KR559508) and MED (GenBank accession numbers MH205753) mtCOI reference sequences.

### 2.4. Haplotype Analysis

Haplotype analysis was carried out using 105 *B. tabaci* (102 MEAM1 and three MED) sequences obtained through 2195Bt and C012/Bt-sh2 primers (GenBank accession numbers MW024919–MW024949, MW025170–MW025179, MW025184–MW025197, MW031122–MW031131, MW046877–MW046891, MW160137–MW160161). Sequences were aligned using MUSCLE in MEGA X [48], and the number of haplotypes were determined based on aligned sequence fragments using DnaSP version 4.10.0 [49]. Minimum spanning haplotype network between *B. tabaci* MEAM1 and *B. tabaci* MED haplotypes was constructed using PopART software [50]. 

### 2.5. Microsatellite Genotyping

For each population, twelve females of *B. tabaci* were genotyped at six loci using the following primers: BEM6, BEM11, BEM15, BEM23, BEM25, and BEM31 [51]. PCR amplification with microsatellite primers was conducted in 12.5 μL reactions composed of 6.25 μL of 2× Type-it Multiplex PCR Master Mix (QIAGEN, Germantown, MD, USA), 1 μL of sterile water, 1.25 μL of forward and reverse primer mix (10 pmol), and 20 ng of DNA template. The forward primers were labeled with the fluorescein derivative 5-carboxyfluorescein (FAM) for microsatellite scoring. The PCR conditions included 94 °C for 7 min, followed by 35 cycles of 1 min at 94 °C, 1 min at 47 °C, 1 min at 73 °C, and a final extension of 72 °C for 1 h. Ready-to-run genotyping reaction solution was generated by mixing 1 μL of PCR products with 9 μL formamide and 1 μL ROX 500 size standard. In 96 well polypropylene micro-titer plates, 1 μL genotyping reaction solution was sent to the Georgia Genomics and Bioinformatics Core (UGA, Athens, GA, USA) for genotyping and amplicon size analysis. In order to estimate the genotyping error rates, duplicate genotypes were generated for each sample across all loci. Alleles were treated as missing data in cases of amplification failure, presence of three or more peaks in the electropherogram, peaks having insufficient height, or mismatches between duplicates. Based on the size of PCR products, two microsatellite markers BEM6 and BEM23 were used for the identification of cryptic species of *B. tabaci* [52]. All genotyped females (n: 12 × 35 = 420) were identified to cryptic species as MEAM1 or MED using these loci. Since only nine out of 420 genotyped females were identified as MED, further population genetic analyses were limited to the 411 MEAM1 genotypes from 35 populations representing 13 host plants and 15 farmscapes (Microsatellite Dataset: https://doi:10.5061/dryad.xgxd254f7).

### 2.6. Genetic Diversity of B. tabaci MEAM1 

Genetic diversity across loci was estimated with several descriptive statistics: number of alleles, evenness of allele frequency, observed heterozygosity, expected heterozygosity, inbreeding coefficient (*F_IS_*), and fixation index (*F_ST_*), using the ‘poppr’ package in R version 3.6.0 [53,54]. Genetic diversity across populations was estimated according to: number of alleles, number of effective alleles, observed heterozygosity, expected heterozygosity, Shannon’s information index and *F_ST_* using GenAlEx6.5 [55]. Whitefly genotypes from each population and locus were assessed for the presence of null alleles, departure from the Hardy–Weinberg equilibrium (HWE), and for linkage disequilibrium between locus pairs using the ‘poppr’ R package. 

Mutation-drift equilibrium among populations was tested using BOTTLENECK v.1.2.02 [56]. The probability of a bottleneck (indicated by significant heterozygote excess) in each population was estimated using a one-tailed Wilcoxon sign-rank test (*p* < 0.05) according to three models: infinite alleles model (IAM), two-phase model (TPM), and stepwise mutation model (SMM) (parameters for TPM: variance = 30.0%, probability = 70.0%, 1000 replications). The probability of a bottleneck was estimated using the Wilcoxon sign-rank test [56]. Signatures of bottlenecks in populations were visually confirmed by examining mode shifts in populations’ allele frequency distributions, where a relative underrepresentation of low-frequency alleles was considered a recent population bottleneck [57]. 

### 2.7. Genetic Differentiation among B. tabaci MEAM1 Populations

Pairwise *F_ST_*, analysis of molecular variance (AMOVA), genetic isolation by distance (Mantel test), STRUCTURE, and principal components analysis (PCA) were used to characterize the population structure and genetic differentiation among collected *B. tabaci* MEAM1 populations. First, pairwise *F_ST_* was calculated using 10,000 bootstrap pseudoreplicates over loci, while accounting for null alleles, using the ‘poppr’ R package. Significance of pairwise *F_ST_* values was estimated with the ‘hierfstat’ R package with 999 permutations [58]. Values of pairwise *F_ST_* < 0.05 were taken as evidence of low differentiation among the populations, while values of *F_ST_* > 0.15 were taken as evidence of high genetic differentiation [29]. Second, AMOVA was performed in R using the ‘poppr’ R package to partition the genetic variance among farmscapes, among host plants within a farmscape, among populations and within populations [53], and significance was tested with 1000 permutations using the *randtest* function in the ‘ade4’ R package [59]. Third, correlation between pairwise genetic distance (*F_ST_*/(1 − *F_ST_*)) and pairwise geographic distance (Ln km) between all pairs of populations was analyzed by Mantel test (9999 permutations) using GenAlEx6.5 [55]. Fourth, Bayesian clustering was implemented in STRUCTURE v2.3.4 [60]. The admixture ancestry model was run with the correlated allele frequency model to calculate the number of distinct genetic clusters (K). STRUCTURE was used to identify the distinct genetic clusters (K) within the dataset by detecting allele frequency differences and assigning the individuals to those clusters based on analysis of likelihoods. The range of possible distinct genetic clusters (K) was set from 1 to 10 with 20 runs for each genetic cluster. Clustering was analyzed with a burn-in period of 50,000 iterations and 1,000,000 Markov Chain Monte Carlo (MCMC) replicates. The most likely number of genetic clusters in the *B. tabaci* MEAM1 populations collected over the farmscape was estimated using log-likelihood values of each K and ΔK in STRUCTURE HARVESTER [57]. Fifth, PCA was implemented in R using ‘adegenet’ and ‘ade4′ R packages [59,61]. PCA was run using the *dudi.pca* function and the first three principal components (PC) encompassing the majority of genetic variability among host plants (PC1: 48.62%, PC2: 24.00%, and PC3: 14.76%) and farmscapes (PC1: 54.34%, PC2: 22.12%, and PC3: 11.06%) were retained. 

## 3. Results

### 3.1. Determination of Whitefly Species

A total of three different whitefly species and two *B. tabaci* cryptic species were observed among the collected populations. The bandedwinged whitefly, *Trialeurodes abutilonea* Haldeman (GenBank accession number MT976143 −99.83% nucleotide similarity with GenBank reference sequence MG817067) and *B. tabaci* MEAM1 (GenBank accession number KR559508 −98.78% nucleotide similarity with GenBank reference sequence LN614546) were present at all collection sites. The greenhouse whitefly, *Trialeurodes vaporariorum* Westwood (GenBank accession number MT976141 −100.00% nucleotide similarity with GenBank reference sequence MK490855) was found on field squash in Spalding county. The citrus whitefly, *Dialeurodes citri* Ashmead (GenBank accession number MT976142 −96.81% nucleotide similarity with GenBank reference sequence JQ340192) was found on horseweed growing in the non-crop vegetation located next to cotton in Mitchell and Sumter counties. *Bemisia tabaci* MED was detected from snap bean, lantana, and eggplant in Clarke, Oconee, and Spalding counties, respectively (GenBank accession number MT976144 −99.75% nucleotide similarity with GenBank reference sequence MH205753). In farms from all three above-stated counties/farmscapes, mixed populations of *B. tabaci* MEAM1 and *B. tabaci* MED were present in the same field (Figure 1). 

### 3.2. Haplotype Analysis 

Analyses of *B. tabaci* mtCOI sequences revealed that out of 105 sequences, 102 were identified as *B. tabaci* MEAM1 and three were identified as *B. tabaci* MED. *Bemisia tabaci* MEAM1 mtCOI sequences had a 98.50 to 100.00% nucleotide identity to the reference sequence (GenBank accession number KR559508). *Bemisia tabaci* MED mtCOI sequences had a 99.50% to 99.75% nucleotide identity to the reference sequence (GenBank accession number MH205753). The mtCOI amino acid sequences of *B. tabaci* MEAM1 and MED were 100.00% identical to their respective reference sequences (MEAM 1: GenBank accession numbers KR559508; MED: GenBank accession number MH205753). Examination of aligned *B. tabaci* mtCOI sequences (690 bp) revealed the presence of one MEAM1 (MEAM1H1) and one MED (MEDH1) haplotype (Figure 2). 

### 3.3. Genetic Diversity of B. tabaci MEAM1

All loci exhibited variation among populations in evenness of allele frequencies (0.43–0.83) and expected heterozygosity (*H**_exp_*) (0.12–0.73), confirming the utility of these microsatellite markers for detecting variability among populations (Table 2). The average frequency of null alleles ranged from 0.021 to 0.12. The number of alleles per locus ranged from 4 to 7 (Table 2). The expected (*H_exp_*) heterozygosity ranged from 0.12 (BEM23) to 0.73 (BEM15) and observed heterozygosity (*H*_o_) ranged from 0.05 (BEM23) to 0.63 (BEM25) (Table 2). *F_IS_*, which describes the difference between observed and expected heterozygosity (*F_IS_* > 0 implies a heterozygote deficit and *F_IS_* < 0 implies heterozygote excess), ranged from −0.16 to 0.51 across the loci. Loci BEM6 and BEM23 had a significantly positive *F_IS_* value, suggesting heterozygote deficits at these loci. *F_ST_* across loci ranged from 0.04 to 0.19 (Table 2). The highest *F_ST_* was observed for the BEM6 marker. No population exhibited evidence of linkage disequilibrium between any loci. 

Genetic diversity observed in *B. tabaci* populations collected from different host plants and farmscapes are shown in Table 3 and Table 4. The mean number of alleles ranged from 2.00 to 4.83 and 2.33 to 4.00 for populations collected from different host plants and farmscapes, respectively (Table 3 and Table 4). The expected (*H_exp_*) and observed heterozygosity (*H*_o_) for populations from different host plants ranged from 0.30 to 0.42 and 0.29 to 0.44, respectively. For populations from different farmscapes, expected (*H_exp_)* and observed heterozygosity (*H*_o_) ranged from 0.32 to 0.49 and 0.14 to 0.47, respectively (Table 4). *F_IS_* for populations collected from different host plants ranged from −0.17 to 0.29 and *F_IS_* for populations collected from different farmscapes ranged from −0.29 to 0.55 (Table 3 and Table 4). Among collections by host plant, significant heterozygote excess was found among whiteflies collected from horseweed and tobacco, and whiteflies collected from eggplant exhibited a significant heterozygote deficit (Table 3). Among collections by farmscape, significant heterozygote excess was found in farmscape 14 and 15 (Wheeler and Montgomery counties, respectively), and significant heterozygote deficits were found in farmscape 3 and 4 (Spalding and Sumter counties, respectively) (Table 4). 

Under all three population genetics models (IAM, TPM, and SMM) significant heterozygote excess was observed among whiteflies collected from okra (Farmscape 3, Spalding) and horseweed (Farmscape 7, Mitchell) (Table 5). However, a mode shift in the allele frequency distribution was only observed for whiteflies collected from horseweed (Table 5). Significant heterozygote excess was additionally detected among populations collected from farmscape 5, but only according to the IAM model. Among farmscapes, mode shifts—indicative of population bottlenecks—were apparent in populations collected from farmscapes 1, 2, 14, and 15 (Table 6). Overall, there was no evidence of widespread recent bottlenecks among *B. tabaci* MEAM1 populations. 

### 3.4. Genetic Differentiation among B. tabaci MEAM1 Populations 

Pairwise *F_ST_* values among *B. tabaci* MEAM1 populations collected from different host plants ranged from 0.01 to 0.05 and pairwise *F_ST_* values among populations collected from different farmscapes ranged from 0.01 to 0.07 (Table 7 and Table 8). Overall, *B. tabaci* MEAM1 populations collected from different host plants and farmscapes exhibited low genetic differentiation. Pairwise *F_ST_* values between populations collected from different host plants or farmscapes were low but were significant between numerous populations (Table 7 and Table 8). The highest significant pairwise *F_ST_* value (0.05) for different hosts was found between tobacco and okra (Table 7) and highest pairwise significant *F_ST_* (0.07) was observed between farmscape 4 and farmscape 15 (Table 8).

The analysis of molecular variance (AMOVA) revealed that most of the genetic variance was partitioned within populations (among and within individuals of a population) (Table 9). The variance partitioned among *B. tabaci* MEAM1 populations from different farmscapes and host plants was 2.00% (Table 9). Overall, AMOVA results suggested that there were no significant genetic differences among populations. Mantel test results revealed no correlation between genetic and geographic distances among populations (*r*^2^ = 0.0008, *p* = 0.429).

Bayesian cluster analysis performed using STRUCTURE identified *K* = 3 as the optimal number of genetic clusters according to log-likehood values of each K and ΔK (Figure 3A,B, Appendix A). However, all populations exhibited a relatively even distribution of ancestry proportions from each genetic cluster, and separation by host plant or farmscape was not observed (Figure 3C, Figure 4). PCA results showed broad overlap among *B. tabaci* individuals along PC1 and PC2, which together accounted for 72.62% and 76.46% of the variation in microsatellite genotypes among whiteflies collected from host plants and farmscapes, respectively. (Figure 5). 

## 4. Discussion

Several factors, such as host plants and local agricultural practices, can influence the genetic diversity and population structure of insect pests inhabiting farmscapes [1,29,62]. The influence of such factors in shaping the population structure and genetics of insect pests such as whiteflies at the farmscape level has been sparsely explored. This study examined the genetic differentiation and structure of *B. tabaci* MEAM1 populations occurring in heterogeneous farmscapes of Georgia, USA. Partial mtCOI gene sequences and six nuclear microsatellite markers were utilized to examine patterns of genetic diversity and differentiation among populations of *B. tabaci* MEAM1. Comparison of partial mtCOI sequences led to the identification of a single predominant *B. tabaci* MEAM1 haplotype occurring throughout the farmscapes of Georgia. Analyses of microsatellite markers further revealed low levels of genetic diversity and differentiation among MEAM1 populations and found no evidence of host-or farmscape-associated differentiation. Overall, results show that a single panmictic population of *B. tabaci* MEAM1 dominates weeds and all crops across the farmscapes that we sampled. In addition to *B. tabaci* MEAM1 and MED cryptic species, three other whitefly species viz., *D. citri*, *T. abutiloneus*, and *T. vaporariorum* were also identified in the collected populations. 

The citrus whitefly, *D. citri* is a serious citrus pest in Florida [63]. However, it is seldom considered a pest in vegetables and row crops in Georgia. Bandedwinged whiteflies, *T. abutiloneus*, and *B. tabaci* MEAM1 were present in all farmscapes, but *B. tabaci* MEAM1 was far more abundant than *T. abutiloneus*. *Trialeurodes*
*abutiloneus* is native and widely distributed throughout United States. Although distributed throughout the farmscapes of Georgia, *T. abutiloneus* rarely reaches numbers that justify treatment with insecticides. The greenhouse whitefly, *T. vaporariorum,* was found on field-grown squash in Spalding county. There is growing evidence that *T. vaporariorum* may not necessarily be limited to greenhouse environments [64,65]. However, in the current study, *T. vaporariorum* was found in just one squash field located near urban landscapes. Therefore, the *T. vaporariorum* individuals that were collected might have dispersed into the squash field from a nearby greenhouse. *Trialeurodes* is the only whitefly genus other than *Bemisia* that has been documented as a plant virus vector [66]. Both *T. abutiloneus* and *T. vaporariorum* are reported vectors of plant viruses in the family *Closteroviridae* [65,66]. *Bemisia tabaci* MED cryptic species was found in snap bean and eggplant fields in Clarke and Spalding counties located in North Georgia, respectively. At both locations, MED individuals were present in the same field as MEAM1 and were limited in number (<15% of the individuals examined for each county). Both locations were in close proximity to urban landscapes; therefore, there is a high likelihood that these isolated MED individuals may have dispersed into these crops from nearby greenhouses or ornamentals. *Bemisia tabaci* MED has a high propensity to develop resistance to insecticides, and its presence in field-grown vegetables can have profound impacts on whitefly management programs [67]. *Bemisia tabaci* MED has replaced *B. tabaci* MEAM1 as the dominant whitefly in certain regions of China [68,69,70]. In the United States, since its documentation ~15 years ago, *B. tabaci* MED has been primarily restricted to ornamentals in greenhouses [67]. Recently, it has also been detected in residential landscapes in Florida [67]. Predicting what will trigger a *B. tabaci* MED outbreak in field crops and vegetables in the United States as in other places is not obvious. As of now, *B. tabaci* MEAM1 seems to be better adapted to the farmscapes in the southeastern United States than *B. tabaci* MED. In a recent study, McKenzie et al., also documented the reoccurrence of New World cryptic species (NW, biotype A) of *B. tabaci* in the United States following its disappearance in the late 1980s [26,71]. Our results provide no evidence for reoccurrence of the indigenous biotype-A within the farmscapes of Georgia; however, we acknowledge that its presence may have gone undetected due to the limited number of samples tested in this study. 

Genetic differentiation (pairwise *F_ST_*) between *B. tabaci* MEAM1 populations collected from different host plants or farmscapes was very low. Likewise, results from population structure analysis (AMOVA, STRUCTURE, and PCA) did not suggest evidence of host- or farmscape-specific genetic clustering. Furthermore, a test of isolation by distance (Mantel test) indicated no correlation between genetic differentiation and geographic distance among all populations. Taken together, results suggest that there could be extensive gene flow among whitefly populations inhabiting various crops and farmscapes aided by frequent wind-aided dispersal and high spatial synchrony among *B. tabaci* populations across farmscapes [72,73]. Low genetic diversity among *B. tabaci* MEAM1 populations observed in the current study could also be influenced by population bottlenecks, founder effects, or high mortality caused by insecticides [34,74,75]. Invasive insects such as *B. tabaci* MEAM1 often experience genetic bottlenecks that can lead to low genetic diversity [74,76]. In the current study, only one out of 13 populations collected from different host plants and one out of 15 populations collected from different farmscapes exhibited evidence of a genetic bottleneck. Overall, there was no substantial evidence for bottleneck effects driving the low genetic differentiation observed among *B. tabaci* MEAM1 populations in Georgia. Nevertheless, the heterozygote excess associated with population bottlenecks is not expected to last more than few generations [77]. This signature could rapidly erode in insects such as *B. tabaci*, which have high reproductive potential and can complete multiple generations (up to 12 in Georgia, United States) within a single calendar year [30]. Thus, signatures associated with earlier population bottleneck effects influencing *B. tabaci* MEAM1 populations since its introduction to the United States in the 1980s might not have been captured in this study. 

*Bemisia tabaci* population genetic analyses carried out at fine spatial scales with no major geographical barriers tend to show no or minimal genetic differences among populations [32,78,79]. However, studies carried out over large geographical areas have reported substantial population structure [31,80]. These studies suggest that *B. tabaci* populations tend to cluster between regions isolated by geographical barriers. Results in this study are in agreement with earlier reports; the low level of genetic diversity observed in the current study might be influenced by lack of geographical barriers between populations. Furthermore, cropping patterns in Georgia might have also contributed to the low genetic differentiation in *B. tabaci* MEAM1 populations. In Georgia, summer cotton is planted within a rotation of spring, fall, and winter vegetable crops [81,82], essentially providing suitable host plants for *B. tabaci* MEAM1 year-around. Cotton is one of the most widely grown crops in Georgia; approximately 1.4 million acres of cotton were planted in 2019 [83]. Widespread availability of susceptible hosts and higher temperatures during the summer allow whiteflies to reproduce extensively, and cotton defoliation could trigger mass dispersal of whiteflies from cotton into fall-planted vegetable crops and also weeds. Over the years, this annual dispersal of whiteflies from cotton to nearby vegetation might have resulted in the genetic uniformity among *B. tabaci* MEAM1 populations across farmscapes. 

Dispersal is a vital component of *B. tabaci* ecology, which not only enables host finding and colonization in constantly changing land cover, but also assists in distribution of favorable genetic traits such as insecticide resistance among populations [73]. In the current study, we did not find genetic differences between whiteflies collected from vegetables (squash, okra, tomato, eggplant, snap beans), row crops (cotton, soybean, tobacco), and weeds (horseweed, lantana), suggesting that whiteflies on vegetables and field crops might regularly disperse from weeds in the vicinity and vice-versa. Cases of insecticide resistance in *B. tabaci* have been well-documented in many parts of the world including in the southeastern United States [84,85,86,87]. Insect growth regulators, diamides, and neonicotinoids are vital classes of insecticides for integrated whitefly management programs in the southeastern United States [67,88,89,90]. A resistance gene arising against these insecticides can quickly disperse into interbreeding populations of *B. tabaci* MEAM1. *Bemisia tabaci* is haplodiploid, wherein the females are diploid, and the males are haploid. Because recessive alleles are always expressed in haploid males, recessive resistance traits can quickly become fixed in populations, especially those with a high ratio of males to females. Occurrence of such rapid fixation of insecticide resistance conferring alleles could essentially jeopardize management programs that rely on applications of insecticides in multiple crops. Knowledge about the genetic uniformity of *B. tabaci* populations over the farmscapes offers an intuitive avenue for slowing the evolution of insecticide resistance and enhancing sustainability in whitefly management. All above mentioned insecticide classes act differently on whiteflies (different modes of action) [91]. Therefore, rotation of these insecticides, along with robust insecticide resistance management programs, would not only slow the resistance evolution but also give a leeway to alter the insecticide application patterns if resistance arises anywhere in the farmscapes. For instance, if high levels of insecticide resistance can be ascribed to a single widespread genetic cluster, then insecticide applications can be adjusted across farmscapes accordingly.

## 5. Conclusions

Whitefly population genetics at broad spatial scales and with respect to invasion routes have been well studied [28,32]. However, less is known about levels of population structure among whitefly populations at finer scales such as farmscapes. Here, we find evidence that whitefly populations occurring in heterogeneous farmscapes comprise a single panmictic population. Such homogeneity among populations could arise from extensive gene flow, though the importance of a recent founder effect cannot be precluded. Extensive gene flow could facilitate the rapid spread of any new trait arising in a local population and warrants further investigation with higher-resolution genetic markers. Results from the current study provide clear evidence for the presence of a single panmictic population over the summer and early fall in Georgia and identify avenues where this information can be used in whitefly management programs. With such low genetic variation within summer and fall populations, one would expect the same *B. tabaci* MEAM1 genetic cluster to prevail and circulate with changing cropping patterns in cooler seasons. However, it remains to be empirically examined.

## Figures and Tables

**Figure 1 insects-11-00834-f001:**
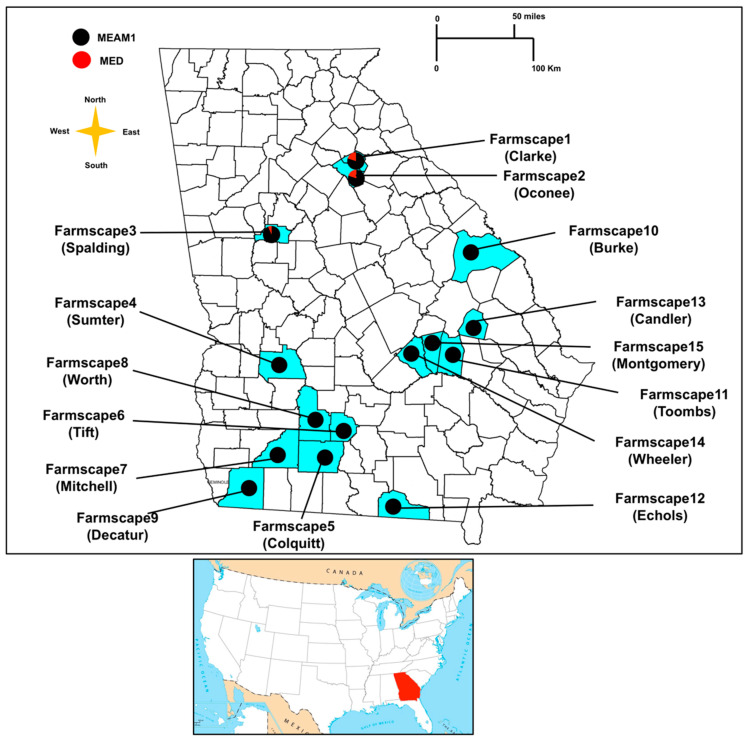
Distribution of *B. tabaci* cryptic species (MEAM1 and MED) in Georgia in 2019 based on mtCOI sequences and microsatellite markers analysis. Pie charts represent the proportion of *B. tabaci* MEAM1 (black) and MED (red) individuals in collected populations.

**Figure 2 insects-11-00834-f002:**
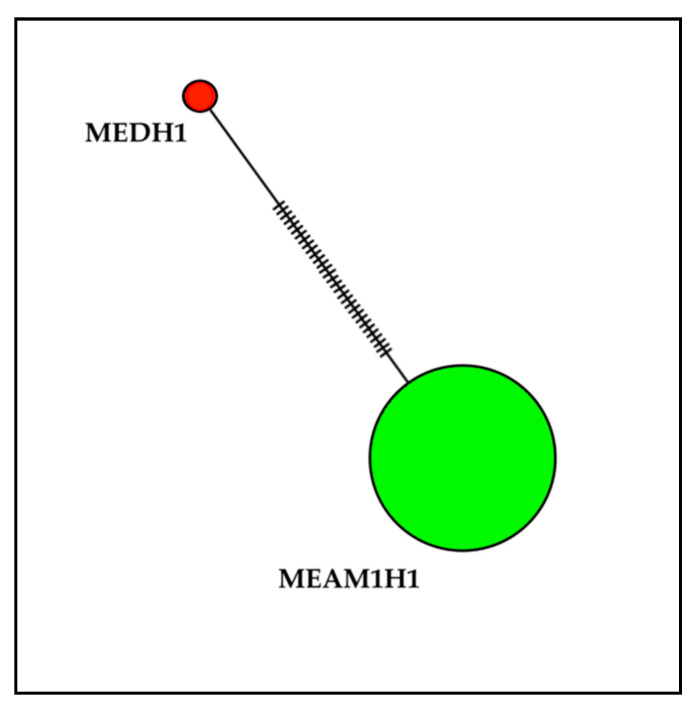
Minimum spanning network of *B. tabaci* haplotypes based on mtCOI sequences. Size of the circles are proportional to the number of individuals in each haplotype. MEDH1 and MEAM1H1 are haplotypes of MED and MEAM1, respectively. Dashed lines between circles represent mutational steps.

**Figure 3 insects-11-00834-f003:**
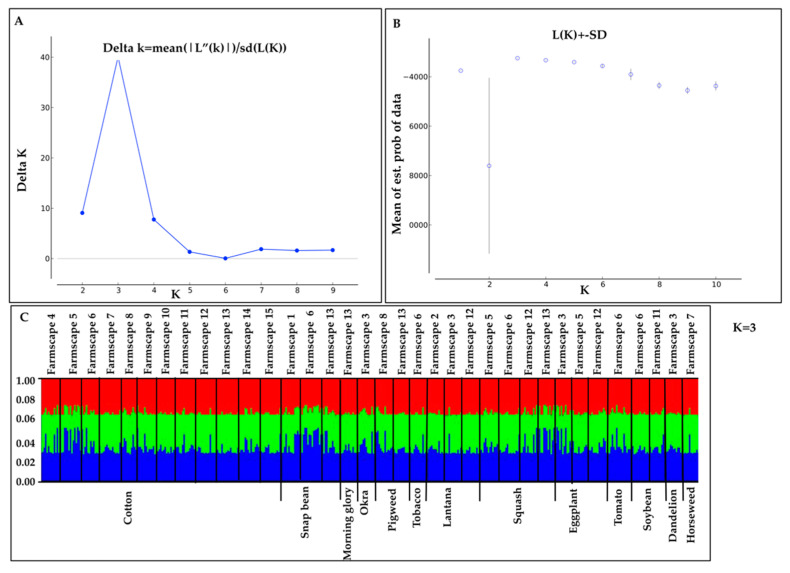
Bayesian clustering analysis results for 35 *B. tabaci* MEAM1 populations based on six microsatellite markers using STRUCTURE v.2.3.2. (**A**) Optimal number of genetic clusters (*K = 3*) following methods described by Evanno et al. 2005. (**B**) Plot of average likelihood L(K) and variance per K. (**C**) Scatter plots at *K* = 3. The length of each line in the bars represents the proportion of the genome in different clusters. Whitefly populations collected from different host plants or farmscapes are separated by a continuous vertical black line.

**Figure 4 insects-11-00834-f004:**
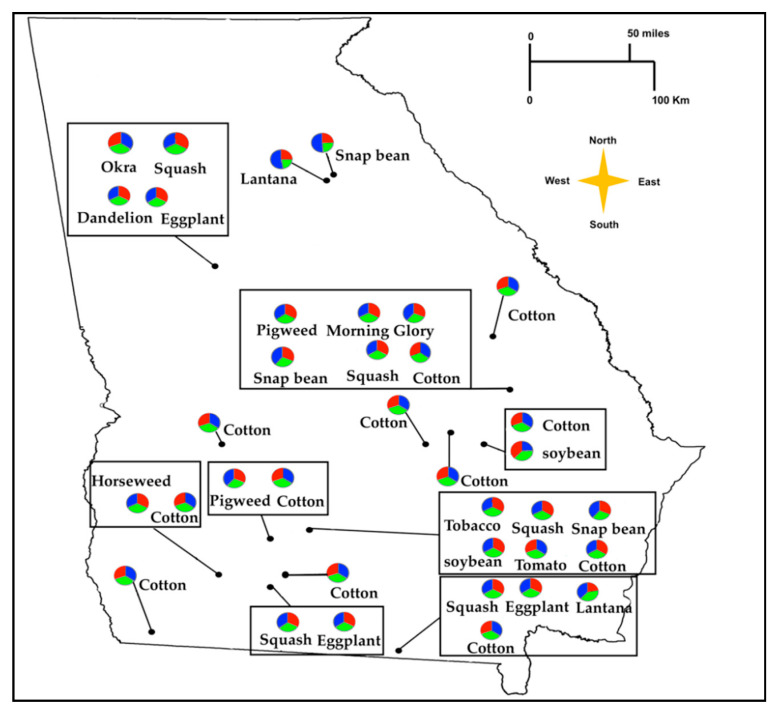
Pie charts showing the proportion of alleles that were inherited from a postulated ancestral population to individuals of 35 *B. tabaci* MEAM1 populations collected from Georgia, USA, using Bayesian clustering implemented in STRUCTURE 2.3.4 at *K =* 3.

**Figure 5 insects-11-00834-f005:**
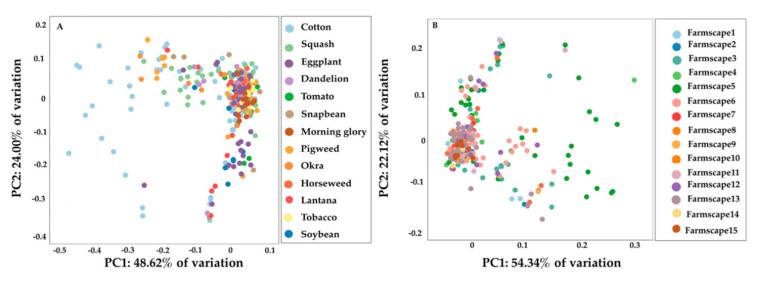
Population structure based on principal component analysis for *B. tabaci* MEAM1 populations collected from different host plants (**A**), and farmscapes (**B**). Dots represent single individuals within the populations. All individuals are clustered together indicating minimum genetic isolation between individuals.

**Table 1 insects-11-00834-t001:** Collection data for whiteflies analyzed in this study.

Population Number	Farmscape	County	Host Plant	Collection Date	GPS Coordinates (DMS) ^a^
1	Farmscape 1	Oconee	Snap bean (*Phaseolus vulgaris* L.)	08/23/2019	33°43’26.5″ N 83°19’41.5″ W
2	Farmscape 2	Clarke	Lantana (*Lantana camara* L.)	08/23/2019	33°54’02.9″ N 83°22’56.6″ W
3	Farmscape 3	Spalding	Okra (*Abelmoschus esculentus* (L.)	07/19/2019	33°15’48.0″ N 84°18’25.5″ W
4	Farmscape 3	Spalding	Dandelion (*Taraxacum officinale* Weber)	07/20/2019	33°15’57.1″ N 84°18’22.4″ W
5	Farmscape 3	Spalding	Eggplant (*Solanum melongena* L.)	07/20/2019	33°15’46.7″ N 84°17’30.6″ W
6	Farmscape 3	Spalding	Squash (*Cucurbita pepo* L.)	07/21/2019	33°15’45.2″ N 84°17’06.8″ W
7	Farmscape 4	Sumter	Cotton (*Gossypium hirsutum* L.)	08/19/2019	32°02’35.2″ N 84°22’13.4″ W
8	Farmscape 5	Colquitt	Cotton (*Gossypium hirsutum* L.)	08/01/2019	31°11’32.0″ N 83°40’18.6″ W
9	Farmscape 5	Colquitt	Squash (*Cucurbita pepo* L.)	08/01/2019	31°12’07.6″ N 83°40’10.8″ W
10	Farmscape 5	Colquitt	Eggplant (*Solanum melongena* L.)	08/01/2019	31°11’23.5″ N 83°43’41.2″ W
11	Farmscape 6	Tift	Snap bean (*Phaseolus vulgaris* L.)	07/23/2019	31°28’17.7″ N 83°31’47.7″ W
12	Farmscape 6	Tift	Squash (*Cucurbita pepo* L.)	07/23/2019	31°29’01.3″ N 83°31’18.3″ W
13	Farmscape 6	Tift	Tomato (*Solanum lycopersicum* Mill)	07/23/2019	31°29’01.3″ N 83°31’18.3″ W
14	Farmscape 6	Tift	Soybean (*Glycine max* Merrill)	07/23/2019	31°29’01.3″ N 83°31’18.3″ W
15	Farmscape 6	Tift	Tobacco (*Nicotiana tabacum* L.)	07/23/2019	31°28’13.0″ N 83°31’54.1″ W
16	Farmscape 6	Tift	Cotton (*Gossypium hirsutum* L.)	08/13/2019	31°30’07.5″ N 83°32’43.0″ W
17	Farmscape 7	Mitchell	Cotton (*Gossypium hirsutum* L.)	08/16/2019	31°16’49.0″ N 84°17’38.1″ W
18	Farmscape 7	Mitchell	Horseweed (*Conyza canadensis* L.)	08/16/2019	31°16’39.8″ N 84°17’54.5″ W
19	Farmscape 8	Worth	Cotton (*Gossypium hirsutum* L.)	08/06/2019	31°35’19.6″ N 83°49’38.4″ W
20	Farmscape 8	Worth	Redroot pigweed (*Amaranthus retroflexus* L.)	08/06/2019	31°35’08.4″ N 83°50’03.3″ W
21	Farmscape 9	Decatur	Cotton (*Gossypium hirsutum* L.)	08/16/2019	30°45’49.7″ N 84°29’09.7″ W
22	Farmscape 10	Burke	Cotton (*Gossypium hirsutum* L.)	08/15/2019	32°52’35.1″ N 82°13’05.2″ W
23	Farmscape 11	Toombs	Soybean (*Glycine max* Merrill)	08/15/2019	32°01’03.2″ N 82°13’15.5″ W
24	Farmscape 11	Toombs	Cotton (*Gossypium hirsutum* L.)	08/15/2019	32°00’55.2″ N 82°13’19.8″ W
25	Farmscape 12	Echols	Cotton (*Gossypium hirsutum* L.)	08/01/2019	30°38’47.1″ N 83°01’42.8″ W
26	Farmscape 12	Echols	Squash (*Cucurbita pepo* L.)	08/01/2019	30°37’43.1″ N 83°02’20.0″ W
27	Farmscape 12	Echols	Eggplant (*Solanum melongena* L.)	08/01/2019	30°39’56.9″ N 83°01’54.9″ W
28	Farmscape 12	Echols	Lantana (*Lantana camara* L.)	08/01/2019	30°39’56.9″ N 83°01’54.9″ W
29	Farmscape 13	Candler	Cotton (*Gossypium hirsutum* L.)	09/02/2019	32°25’44.0″ N 82°04’50.7″ W
30	Farmscape 13	Candler	Snap bean (*Phaseolus vulgaris* L.)	09/02/2019	32°25’52.5″ N 82°04’27.5″ W
31	Farmscape 13	Candler	Squash (*Cucurbita pepo* L.)	09/02/2019	32°26’03.6″ N 82°03’59.1″ W
32	Farmscape13	Candler	Redroot pigweed (*Amaranthus retroflexus* L.)	09/02/2019	32°26’03.6″ N 82°03’59.1″ W
33	Farmscape 13	Candler	Purple morning glory (*Ipomoea purpurea* L.)	09/02/2019	32°26’15.7″ N 82°03’52.4″ W
34	Farmscape 14	Wheeler	Cotton (*Gossypium hirsutum* L.)	08/21/2019	32°06’12.3″ N 82°48’21.6″ W
35	Farmscape 15	Montgomery	Cotton (*Gossypium hirsutum* L.)	08/21/2019	32°11’59.7″ N 82°30’12.6″ W

**^a^** Coordinates are in the degrees, minutes, seconds format (DMS).

**Table 2 insects-11-00834-t002:** Genetic diversity of thirty-five *B. tabaci* populations across six microsatellite markers.

Locus	Number of Alleles	Evenness	Expected Heterozygosity (*H_exp_*)	Observed Heterozygosity (*H*_o_)	Inbreeding Coefficient (*F_IS_*)	Fixation Index (*F_ST_*)
**BEM6**	5.00	0.43	0.14	0.08	0.36	0.19
**BEM11**	7.00	0.74	0.55	0.61	−0.16	0.08
**BEM15**	7.00	0.83	0.73	0.59	0.14	0.05
**BEM23**	5.00	0.44	0.12	0.05	0.51	0.10
**BEM25**	7.00	0.72	0.60	0.63	0.03	0.04
**BEM31**	4.00	0.43	0.15	0.10	0.33	0.18
**Mean**	5.80	0.60	0.38	0.34	0.12	0.11

**Table 3 insects-11-00834-t003:** Genetic diversity of *B. tabaci* MEAM1 populations collected from different plant species based on six microsatellites markers.

Population	Sample Size	Mean number of Alleles	Shannon’s Information Index (I)	Expected Heterozygosity (*H_exp_*)	Observed Heterozygosity (*H*_o_)	Inbreeding Coefficient (*F_IS_*) ^1^
**Cotton**	144	4.83	0.80	0.41	0.36	0.23
**Soybean**	24	3.17	0.72	0.35	0.37	−0.08
**Squash**	60	4.33	0.72	0.37	0.38	−0.04
**Tomato**	12	2.83	0.66	0.33	0.29	0.08
**Snap bean**	36	3.50	0.66	0.35	0.41	0.10
**Lantana**	24	3.33	0.74	0.38	0.35	0.25
**Horseweed**	10	2.83	0.70	0.35	0.40	**−0.15**
**Pigweed**	24	3.17	0.71	0.40	0.31	0.22
**Okra**	12	2.00	0.53	0.39	0.32	0.21
**Dandelion**	12	2.50	0.61	0.38	0.44	0.01
**Eggplant**	36	4.17	0.86	0.42	0.32	**0.29**
**Tobacco**	12	2.67	0.57	0.31	0.37	**−0.17**
**Morning glory**	12	2.50	0.55	0.30	0.29	0.01
**Mean**	32.15	3.22	0.68	0.37	0.30	0.10

^1^ Numbers indicated in bold font and underlined are significantly different from zero. Significant *F_IS_* indicates that populations are not mating randomly.

**Table 4 insects-11-00834-t004:** Genetic diversity of *B. tabaci* MEAM1 populations collected from different farmscapes based on six microsatellites markers.

Population	Sample Size	Mean No. of Alleles	Shannon’s Information Index (I)	Expected Heterozygosity (*H_exp_*)	Observed Heterozygosity (*H*_o_)	Inbreeding Coefficient (*F_IS_*) ^1^
**Farmscape1**	12	2.83	0.70	0.39	0.36	0.23
**Farmscape2**	12	2.83	0.72	0.39	0.38	0.09
**Farmscape3**	48	3.33	0.80	0.41	0.32	**0.33**
**Farmscape4**	12	2.67	0.70	0.34	0.14	**0.55**
**Farmscape5**	36	4.00	0.88	0.49	0.47	0.07
**Farmscape6**	72	3.83	0.82	0.37	0.36	−0.02
**Farmscape7**	22	3.17	0.79	0.34	0.36	−0.06
**Farmscape8**	24	2.83	0.64	0.37	0.37	0.24
**Farmscape9**	12	2.83	0.74	0.35	0.39	0.08
**Farmscape10**	12	2.83	0.61	0.32	0.33	−0.08
**Farmscape11**	24	2.83	0.57	0.35	0.35	0.13
**Farmscape12**	48	3.33	0.65	0.34	0.35	0.03
**Farmscape13**	60	3.83	0.68	0.35	0.35	0.11
**Farmscape14**	12	2.33	0.54	0.33	0.43	**−0.29**
**Farmscape15**	12	2.33	0.61	0.33	0.42	**−0.26**
**Mean**	27.87	3.05	0.69	0.39	0.36	0.23

^1^ Numbers indicated in bold font and underlined are significantly different from zero. Significant *F_IS_* indicates that populations are not mating randomly.

**Table 5 insects-11-00834-t005:** Wilcoxon signed-rank test for mutation-drift equilibrium for 35 *B. tabaci* MEAM1 populations collected from different host plants, based on six microsatellite loci.

Wilcoxon Test *p*-Values ^1^				
	Infinite Alleles Model IAM	Two-Phase Model TPM	Stepwise Mutation Model SMM	
Host Plants	Heterozygosity Excess	Heterozygosity Excess	Heterozygosity Excess	Mode Shift
**Cotton**	0.50	0.78	0.99	L
**Soybean**	0.31	0.50	0.68	L
**Squash**	0.71	0.98	1.00	L
**Tomato**	0.56	0.93	0.96	L
**Snap bean**	0.68	0.68	0.96	L
**Lantana**	0.31	0.50	0.89	L
**Horseweed**	0.06	0.06	0.06	S
**Pigweed**	0.31	0.31	0.40	L
**Okra**	**0.03**	**0.03**	**0.03**	L
**Dandelion**	0.31	0.68	1.00	L
**Eggplant**	0.40	0.68	1.00	L
**Tobacco**	0.06	0.12	0.81	L
**Morning glory**	0.13	0.13	0.81	L

^1^ Numbers indicated in bold font and underlined are significant at *p* < 0.05; L: normal L-shaped distribution; S: shifted mode distribution.

**Table 6 insects-11-00834-t006:** Wilcoxon signed-rank test for mutation-drift equilibrium for 35 *B. tabaci* MEAM1 populations collected from different farmscapes, based on six microsatellite loci.

Wilcoxon Test *p*-Values ^1^				
	Infinite Alleles Model IAM	Two-Phase Model TPM	Stepwise Mutation Model SMM	
Farmscapes	Heterozygosity Excess	Heterozygosity Excess	Heterozygosity Excess	Mode Shift
**Farmscape1**	0.92	0.40	0.89	S
**Farmscape2**	0.31	0.41	0.41	S
**Farmscape3**	0.05	0.31	0.41	L
**Farmscape4**	0.56	0.84	0.94	L
**Farmscape5**	**0.04**	0.42	0.96	L
**Farmscape6**	0.78	0.57	0.98	L
**Farmscape7**	0.63	0.63	0.63	L
**Farmscape8**	0.05	0.40	0.59	L
**Farmscape9**	0.84	0.84	1.00	L
**Farmscape10**	0.16	0.16	0.56	L
**Farmscape11**	0.41	0.69	0.92	L
**Farmscape12**	0.58	0.92	0.98	L
**Farmscape13**	0.59	0.92	0.98	L
**Farmscape14**	0.06	0.06	0.06	S
**Farmscape15**	0.06	0.06	0.06	S

^1^ Numbers indicated in bold font and underlined are significant at *p* < 0.05; L: normal L-shaped distribution; S: shifted mode distribution.

**Table 7 insects-11-00834-t007:** Pairwise *F_ST_* values among *B. tabaci* MEAM1 populations collected from different host plants based on six microsatellites markers.

	Cotton	Soybean	Squash	Tomato	Snapbean	Lantana	Horseweed	Pigweed	Okra	Dandelion	Eggplant	Tobacco
**Soybean**	0.01											
**Squash**	0.01	0.02										
**Tomato**	0.01	0.03	0.02									
**Snapbean**	0.01	0.02	0.02	0.01								
**Lantana**	0.01	0.02	0.02	0.02	0.01							
**Horseweed**	0.01	**0.03**	0.02	0.02	0.02	0.02						
**Pigweed**	0.01	**0.03**	0.02	**0.03**	0.01	0.02	0.03					
**Okra**	0.01	**0.03**	0.02	**0.03**	0.02	0.02	**0.04**	0.01				
**Dandelion**	0.00	0.02	0.01	**0.04**	0.02	0.01	0.03	0.02	0.03			
**Eggplant**	0.01	0.02	0.02	0.02	0.02	0.01	0.02	0.02	0.03	0.01		
**Tobacco**	0.01	0.02	0.01	**0.03**	0.01	0.02	0.02	0.02	**0.05**	**0.04**	0.02	
**Morning Glory**	0.01	0.02	0.02	0.03	0.01	0.02	0.04	0.02	**0.04**	0.03	0.01	0.01

Pairwise *F_ST_* values in bold font and underlined are significant at *p* < 0.05.

**Table 8 insects-11-00834-t008:** Pairwise *F_ST_* values among *B. tabaci* MEAM1 populations collected from different farmscapes based on six microsatellites markers.

	Farmscape 1	Farmscape 2	Farmscape 3	Farmscape 4	Farmscape 5	Farmscape 6	Farmscape 7	Farmscape 8	Farmscape 9	Farmscape 10	Farmscape 11	Farmscape 12	Farmscape 13	Farmscape 14
**Farmscape 2**	**0.04**													
**Farmscape 3**	0.02	**0.01**												
**Farmscape 4**	**0.04**	**0.04**	0.02											
**Farmscape 5**	0.02	0.03	0.02	0.03										
**Farmscape 6**	0.01	0.01	0.01	0.01	0.02									
**Farmscape 7**	**0.03**	**0.03**	0.01	0.03	0.02	0.03								
**Farmscape 8**	0.02	0.02	0.01	0.02	0.01	0.01	0.01							
**Farmscape 9**	**0.03**	**0.04**	0.02	**0.05**	0.02	0.01	0.01	0.01						
**Farmscape 10**	**0.03**	**0.05**	0.02	0.02	0.02	0.01	0.02	0.02	0.03					
**Farmscape 11**	0.03	0.03	0.01	0.03	0.02	0.01	0.02	0.01	0.02	0.01				
**Farmscape 12**	0.02	0.02	0.01	0.01	0.02	0.01	0.01	0.01	0.01	0.01	0.01			
**Farmscape 13**	0.01	0.02	0.02	0.01	0.02	0.01	0.01	0.01	0.01	0.01	0.01	0.01		
**Farmscape 14**	**0.05**	**0.05**	0.02	**0.05**	0.02	0.01	0.01	0.03	0.03	0.03	0.02	0.01	0.01	
**Farmscape 15**	**0.06**	**0.06**	0.03	**0.07**	0.02	0.01	0.02	0.04	0.04	**0.06**	**0.05**	0.03	0.02	0.02

Pairwise *F_ST_* values in bold font and underlined are significant at *p* < 0.05

**Table 9 insects-11-00834-t009:** Hierarchical analysis of molecular variance (AMOVA) for the 35 *B. tabaci* MEAM1 populations collected from Georgia, USA, based on six microsatellite markers. (**A**) Among populations collected from different host plants. (**B**) Among populations collected from different farmscapes.

Source of Variation	Degrees of Freedom	Sums of Squares	Mean Sums of Squares	% Variation	*p*-Value
**A, Host Plants**					
Among host plants	12	34.35	2.86	2.00	0.39
Among populations within a host plant	22	36.51	1.66	1.00	0.32
Among individuals within a population	409	708.27	1.64	23.00	<0.001
Within individuals	444	442.00	1.00	74.00	<0.001
Total	887	1184.63	1.34	100.00	
**B, Farmscapes**					
Among farmscapes	14	45.88	3.28	2.00	0.43
Among populations within a farmscape	20	40.42	2.01	0.00	0.63
Among individuals within a population	409	680.32	1.59	22.00	<0.001
Within individuals	444	446.50	1.01	76.00	<0.001
Total	887	1172.69	1.32	100.00	

Significance at *p* < 0.01 based on 999 permutation.

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
