# Peer review of "Low Genetic Variability in *Bemisia tabaci* MEAM1 Populations within Farmscapes of Georgia, USA"

_insects, 2020, doi:10.3390/insects11120834_

Round 1

Reviewer 1 Report

Title: “Low Genetic Variability in Bemisia tabaci MEAM1 Populations within Farmscapes of Georgia, USA”

Authors: Gautam et al.

The manuscript examines the effect of host plants and farmscapes in the genetic structure of B. tabaci populations in Georgia. The manuscript is quite interesting, well written and provides novel information, which has also practical importance for the management of the pest. Both mitochondrial and microsatellite DNA markers have been applied and in depth statistical analysis has been performed.

I like very much the manuscript and I have no comments apart that the number of the tables could be reduced and some of them could be provided as supplementary material. However, this is only a suggestion. Therefore, I recommend that the manuscript should be accepted for publication.

Author Response

Our responses in bold

Title: “Low Genetic Variability in Bemisia tabaci MEAM1 Populations within Farmscapes of Georgia, USA”

Authors: Gautam et al.

The manuscript examines the effect of host plants and farmscapes in the genetic structure of B. tabaci populations in Georgia. The manuscript is quite interesting, well written and provides novel information, which has also practical importance for the management of the pest. Both mitochondrial and microsatellite DNA markers have been applied and in depth statistical analysis has been performed.

I like very much the manuscript and I have no comments apart that the number of the tables could be reduced and some of them could be provided as supplementary material. However, this is only a suggestion. Therefore, I recommend that the manuscript should be accepted for publication.

We appreciate your kind suggestions. Reviewers’ suggested changes to several tables, those changes have now been incorporated and the tables modified accordingly.

Reviewer 2 Report

In this manuscript, population structure was analyzed in 35 populations of whitefly, Bemisia tabaci MEAM1 collected from 15 farmscapes in Georgia.  The study region comprised 14 species of both crop and non-crop plants, and population genetics were assessed utilizing segments of mtCOI and 6 microsatellite loci as molecular markers.  The paper is well-written, easy to follow and interesting, and the conclusions of low genetic diversity, lack of population structure and high gene flow were convincingly supported by the analyses presented.

My only major criticism is that the need for phylogenetic analysis (Figure 2) in this study is not at all clear, given that all of the 102 MEAMI 3' COI sequences used to construct the tree consisted of a single haplotype.  The different MEAMI branch lengths in the 3-dimensional tree give the impression of high genetic variability, although there is only a single haplotype.  I strongly suggest deleting the tree, and possibly inserting a haplotype network, which would also show the number of mutational steps separating the MEAMI and MED haplotypes.

Minor comments:

line 119.  Because the B. tabaci MED outgroup is not readily apparent in Table 1 amongst the 35 populations of MEAM1, suggest replacing "whiteflies" with "B. tabaci MEAM1", and then inserting "(Population No. 7)' following B. tabaci MED on line 125 for clarity.

line 142.  I believe "prime" instead of "primer" was intended here, but even the word "prime" is repetitious and not necessary following 5'. 

line 158.  Delete "primer" following 3'.

line 169.  Because sample sizes of 102 and 105 are both used in this paper when referring to B. tabaci, recommend clarifying this here by inserting "(MEAMI plus MED)" following B. tabaci.

line 175.  Delete the period following "Bemisia"

Table 7, and Figures 3–5.   Font sizes for data entries are quite small and need to be increased substantially.

Author Response

our responses in bold:

My only major criticism is that the need for phylogenetic analysis (Figure 2) in this study is not at all clear, given that all of the 102 MEAMI 3' COI sequences used to construct the tree consisted of a single haplotype.  The different MEAMI branch lengths in the 3-dimensional tree give the impression of high genetic variability, although there is only a single haplotype.  I strongly suggest deleting the tree, and possibly inserting a haplotype network, which would also show the number of mutational steps separating the MEAMI and MED haplotypes.

As suggested phylogenetic analysis has now been removed and replaced with a haplotype network showing number of mutational steps separating the MEAMI and MED haplotypes. Line 270.

Minor comments:

line 119.  Because the B. tabaci MED outgroup is not readily apparent in Table 1 amongst the 35 populations of MEAM1, suggest replacing "whiteflies" with "B. tabaci MEAM1", and then inserting "(Population No. 7)' following B. tabaci MED on line 125 for clarity.

Species identification of the collections indicated the presence of three other species besides B. tabaci and MEAM1 and MED B. tabaci cryptic species. Hence the usage of the generic term ‘whiteflies’. The table has been modified as well since the phylogenetic tree was removed.

line 142.  I believe "prime" instead of "primer" was intended here, but even the word "prime" is repetitious and not necessary following 5'. 

Text edited as suggested. Line 139.

line 158.  Delete "primer" following 3'.

Text edited as suggested. Line 156.

line 169.  Because sample sizes of 102 and 105 are both used in this paper when referring to B. tabaci, recommend clarifying this here by inserting "(MEAMI plus MED)" following B. tabaci.

Sentence has been rephrased for clarity. Line: 169

line 175.  Delete the period following "Bemisia"

As suggested by another reviewer, sentence has been removed from the text.

Table 7, and Figures 3–5.   Font sizes for data entries are quite small and need to be increased substantially.

The font sizes have now been increased for more clarity.